# Impact of PET/CT for Assessing Response to Immunotherapy—A Clinical Perspective

**DOI:** 10.3390/jcm9113483

**Published:** 2020-10-28

**Authors:** David Lang, Gerald Wahl, Nikolaus Poier, Sebastian Graf, David Kiesl, Bernd Lamprecht, Michael Gabriel

**Affiliations:** 1Department of Pulmonology Johannes Kepler University Hospital Linz, Krankenhausstrasse 9, 4020 Linz, Austria; david.lang@kepleruniklinikum.at (D.L.); bernd.lamprecht@kepleruniklinikum.at (B.L.); 2Department of Dermatology and Venerology, Johannes Kepler University Hospital Linz, Krankenhausstrasse 9, 4020 Linz, Austria; gerald.wahl@kepleruniklinikum.at; 3Department of Otorhinolaryngology, Head and Neck Surgery, Johannes Kepler University Hospital Linz, Krankenhausstrasse 9, 4020 Linz, Austria; nikolaus.poier@kepleruniklinikum.at; 4Department of Urology and Andrology, Johannes Kepler University Hospital Linz Krankenhausstrasse 9, 4020 Linz, Austria; sebastian.graf@kepleruniklinikum.at; 5University Clinic of Hematology and Internal Oncology Johannes Kepler University Hospital Linz, Krankenhausstrasse 9, 4020 Linz, Austria; david.kiesl@kepleruniklinikum.at; 6Institute of Nuclear Medicine and Endocrinology, Johannes Kepler University Hospital Linz, Krankenhausstrasse 9, 4020 Linz, Austria

**Keywords:** immune checkpoint inhibitor, non-small cell lung cancer, melanoma, lymphoma, head and neck cancer, renal cancer, immunotherapy-related adverse event, pseudoprogression, tracer, CAR-T

## Abstract

Cancer immunotherapy using immune-checkpoint inhibitors (ICI) has revolutionized the therapeutic landscape of various malignancies like non-small-cell lung cancer or melanoma. Pre-therapy response prediction and assessment during ICI treatment is challenging due to the lack of reliable biomarkers and the possibility of atypical radiological response patterns. Positron emission tomography/computed tomography (PET/CT) enables the visualization and quantification of metabolic lesion activity additional to conventional CT imaging. Various biomarkers derived from PET/CT have been reported as predictors for response to ICI and may aid to overcome the challenges clinicians currently face in the management of ICI-treated patients. In this narrative review, experts in nuclear medicine, thoracic oncology, dermatooncology, hemato- and internal oncology, urological and head/neck tumors performed literature reviews in their respective field and a joint discussion on the use of PET/CT in the context of ICI treatment. The aims were to give a clinical overview on present standards and evidence, to identify current challenges and fields of research and to enable an outlook to future developments and their possible implications. Multiple promising studies concerning ICI response assessment or prediction using biomarkers derived from PET/CT alone or as composite biomarkers have been identified for various malignancies and disease stages. Of interest, additional major incentives in the field may evolve from novel tracers specifically targeting immune-checkpoint molecules which could allow not only response assessment and prognosis, but also visualization of histological tumor cell properties like programmed death-ligand (PD-L1) expression in vivo. Despite the broad range of existing literature on PET/CT-derived biomarkers in ICI therapy, implications for daily clinical practice remain elusive. High-quality prospective data are urgently warranted to determine whether patients benefit from the application of PET/CT in terms of prognosis. At the moment, the lack of such evidence as well as the absence of standardized imaging methods and biomarkers still precludes PET/CT imaging to be included in the relevant clinical practice guidelines.

## 1. Introduction

Positron emission tomography/computed tomography (PET/CT) constitutes a major progress in oncology imaging, as it augments CT with the additional dimension of metabolic activity. Primarily used in staging and to some extent in response assessment of various malignancies, research for additional applications of PET/CT is currently evolving towards prognosis estimation and prediction of response to certain therapies, especially in the field of immunotherapy [1]. 

Cancer immunotherapy through immune-checkpoint inhibitors (ICI) has revolutionized the world of medical oncology, achieving major and long-term treatment responses also in metastatic disease that would have been unthinkable only a few years ago. Fields of application of ICI therapies have rapidly expanded to different tumor entities and a multitude of ICI substances in various treatment regimens has subsequently become available [2]. Still, by far, not every patient responds to ICI therapies, and existing biomarkers do not allow to predict response precisely on an individual patient’s level. Thus, prediction and monitoring of response to ICI treatment has become a major research target in the recent years.

As the term “immunotherapy” is rather heterogenous and applies to a multitude of different antineoplastic therapies, this review aimed to give a focused clinical perspective on the current application and research advances of PET/CT, especially in the context of ICI therapy. Furthermore, although other newly developed tracer substances beyond ^18^F-FDG are discussed in this article, the term “PET/CT” is used synonymously for ^18^F-FDG PET/CT, unless otherwise specified.

## 2. Methods

Checkpoint inhibitor therapy itself, as well as the value of PET/CT in different malignancies and disease stages, are highly heterogenous. There is a general lack of high-quality large-scale trials and prospective randomized data for PET/CT-based response evaluation in ICI-treated patients. Although countless studies in that field have been reported, evidence is mostly derived from retrospective study settings with low patient numbers. Therefore, up to now, there is no standard of care (SOC) regarding ICI therapy assessment using PET/CT.

Thus, we chose a structured clinical approach to this review: Experts in the fields of nuclear medicine, thoracic oncology, internal- and hemato-oncology, dermatological, urological and head and neck tumors conducted individual literature reviews based on their respective relevant current clinical practice guidelines as well as on literature research based on MEDLINE using PubMed search of articles within the last 5 years. Each expert was then asked to:(a)Give an overview of the current applications of PET/CT in the context of ICI therapy in current clinical practice within the respective field.(b)Identify up-to-date studies on PET/CT and ICI therapy that might be relevant to the future clinical approaches in the field.(c)Identify current challenges, perspectives, and possible research targets concerning PET/CT and assessment of ICI therapy response within the respective clinical field.

Because of the lack of larger prospective trials, the inclusion of relevant literature into this narrative review, coordinated by M.G. and D.L., was based on the individual decision of each expert with regard to their specific perspective (D.L. and B.L., Pulmonology; G.W., Dermatology; N.P., Otorhinolaryngology; S.G., Urology; D.K., Hematology and Internal Oncology).

D.L. accomplished the compilation of the manuscript, which was then critically revised and finally approved by all other authors.

As mentioned above, this review article mainly refers to the application of PET/CT imaging in patients undergoing immunotherapy with programmed cell death protein 1 (PD-1)/PD-L1 or cytotoxic T-lymphocyte-associated protein 4 (CTLA-4)-directed immune checkpoint inhibitors (ICI), while other targeted anticancer therapies, sometimes also referred to as “immunotherapy”, e.g., antibodies against other tumor-specific antigens or tyrosine kinase inhibitors, are not discussed. However, our approach also includes a brief section on PET/CT and bi-specific T-Cell Engagers (BiTEs) as well as Chimeric Antigen Receptor (CAR-T) cell therapy in the hemato-oncology section of the article. 

## 3. PET/CT in Immune-Checkpoint Inhibitor Therapy—Technical Aspects, Application and Research

### 3.1. Technical Aspects of PET/CT in Assessing Immunotherapy Response

The advent of ICI in the treatment of various malignant tumors constitutes a milestone in clinical oncology. However, the complex immunological response to these novel agents involving the tumor microenvironment has only been incompletely understood and poses a major challenge in molecular imaging strategies.

Accurate assessment of response to antineoplastic therapy is essential to recognize treatment failure at an early stage and to be able to adapt therapy timely. Classically, this would be indicated by an increase in number and/or size of tumor lesions [3]. Using positron emission tomography (PET), also the change in ^18^F-FDG uptake in malignant lesions as quantified by the standardized uptake value (SUV) can be used as a biomarker for therapy response. This metabolic biomarker dimension has been included in many oncological treatment concepts. A summary of quantitative biomarkers derived from PET/CT imaging that can be easily (semi)-automatically determined is shown in Table 1.

In the context of ICI therapy, however, the efficacy of quantitative measurement of ^18^F-FDG uptake may be diminished and sometimes misleading. Enhanced ^18^F-FDG uptake can also be triggered by the activation of the tumor microenvironment with an increased influx and activity of immune cells like T-lymphocytes induced by ICI therapy itself [13,14]. In contrast to aerobic glycolysis in highly differentiated tissues, the so-called Warburg effect in more proliferative neoplastic tissues leads to an increase in anaerobic glycolysis, and thus to an increase in general glucose turnover [15]. At the same time, during immunotherapy, anti-PD-1 activation also stimulates the tumor microenvironment and consequently upregulates glucose transporter (GLUT) mRNA and GLUT proteins, leading to increased glucose consumption as a result of the immunological anti-tumor reaction [16,17]. This altered metabolic situation can conceal the actual treatment response and, under certain circumstances, even lead to false positive scan results. Therefore, new approaches to PET/CT assessment in patients receiving ICI therapy are required [18]. 

### 3.2. Standardizing Response-Assessment in PET/CT Imaging

Basic evaluation of therapy response using PET/CT can be accomplished using qualitative parameters, like the decrease or increase in metabolic of active tumor lesions. Such binary “good or poor” categorization is rather robust and can be used, e.g., for end-of-treatment assessment. Particularly complete metabolic response in PET represents an important individual decision-making criterion and generally indicates a favorable long-term outcome [19]. An example of such positive sustained response is shown in Figure 1. Similarly, the appearance of new lesions in follow-up is of higher clinical relevance than changes in preexisting lesions [20].

However, an such approach does not allow a fine-tuned therapy assessment and is not suitable for clinical trials, where reproducible and standardized response data are warranted. In conventional CT, this led to the introduction of standardized response evaluation criteria in solid tumors (RECIST) using the classifications complete/partial remission (CR/PR), stable disease (SD) and progressive disease (PD) [21]. Concerning PET/CT, analogous criteria have been suggested, such as the “European Organization for Research and Treatment of Cancer (EORTC) PET Criteria” or more recently the “PET response criteria in solid tumors” (PERCIST) [5,22]. The aim of these assessment tools is to improve the clinical value of metabolic therapy assessment in terms of accuracy, reproducibility and to enable earlier response prediction as compared to conventional imaging techniques, e.g., by CT using size parameters [5]. 

One major point that needs to be considered in that context is that quantitative indices (including textural analysis) are dependent on the quality of the PET images. Results may vary distinctly between up-to-date imaging technology and equipment ten or twenty years old. Also, comparability is usually impaired when different reconstruction methodologies are used. Both can pose difficulties especially when comparing quantitative measures between institutions [23,24].

### 3.3. PET/CT and Atypical Response Patterns in Immune-Checkpoint Inhibitor Therapy

In ICI therapy, atypical response patterns pose additional challenges to classical radiological as well as PET-based response evaluations. 

One such feature is pseudoprogression [25], which mimics disease progression, although it actually represents a hypermetabolic “flare-phenomenon” caused by the initial T-cell tumor infiltration. Frequently, such patients witness an objective response in the further course of therapy. Pseudoprogression is found considerably less frequently than effective disease progression, with reported rates below 10% [18,25]. 

Dissociated response is a similar atypical response pattern, with a mixture of lesions responding and progressing simultaneously [26,27]. Its reported frequency varies but might be similar to pseudoprogression, and prognosis is more favorable than for PD [26]; therefore, continuation of ICI treatment should be considered in such patients [27].

Durable responses to ICI therapy are not comprehensively defined but can occur both after primary partial or complete remission or out of stable disease [25]. As shown in Figure 1, 18F-FDG PET/CT can be used to determine long-term remission also on the metabolic level.

An unfavorable feature of atypical response is hyperprogression, which denotes patients presenting with an accelerated tumor growth rate early after ICI initiation, occurring in up to 7% [18,28]. It results in a very poor prognosis and is associated with widespread metastatic disease in the majority of cases [18,29]. Hyperprogression can be easily discovered by CT as well as by ^18^F-FDG PET-CT, using quantitative parameters, e.g., total lesion glycolysis (TLG) [29].

To account for these imaging challenges associated with atypical response patterns, RECIST criteria have been expanded to iRECIST, which is now commonly used for CT re-staging in ICI patients [30]. Analogously, several similar approaches have been suggested for PET/CT, such as iPERCIST [31], combining RECIST and PERCIST with the introduction of the response category “unconfirmed progressive metabolic disease”. Several other models, such as PERCRIT [32], PERCIMT [13], imPERCIST5 [33] (all for melanoma) or LYRIC (for Hodgkin lymphoma) [34] have been published. However, none of these criteria have been prospectively validated, and all are based on comparably small patient cohorts. Furthermore, differences in timing schedule, reference standards, tumor entities, outcome measures and response classification still limit the practicability of these response criteria in clinical routine.

### 3.4. PET/CT for Immunotherapy-Related Adverse Events

Immune-related adverse events (IRAE) such as thyreoiditis, pneumonitis or hypophysitis may lead to unusual patterns of ^18^F-FDG tracer uptake in various involved organs. These therapy-related findings represent possible pitfalls in PET/CT interpretation, especially as such side effects may not necessarily be clinically evident [35,36,37]. However, they can also have prognostic implications, as their occurrence may indicate a more favorable prognosis [38,39,40].

### 3.5. Novel Approaches to PET/CT Imaging and Tracers beyond ^18^F-FDG

A novel approach for baseline PET/CT imaging and prognosis estimation apart from traditional staging and response criteria refers to textural features within the tumor lesions. Texture analysis means a systematic, computer-aided evaluation of image data with special regards to features like heterogeneity within one or between individual tumor lesions. Such biomarkers have been correlated with clinical outcome parameters and may pose a major step towards a truly personalized therapy approach. Promising first results have been reported in patients with malignant melanoma treated with vemurafenib and ipilimumab [41]. For lung cancer, similar data could be shown considering the fraction of necrosis within the tumor lesions, which was also linked to the presence of CD8-positive lymphocytes in histological samples [42]. 

As already discussed, a general problem concerning ^18^F-FDG as a tracer for cancer treated with ICI is that therapy itself leads to an influx of inflammatory cells leading to enhanced PET activity [43]. Thus, more specific radiopharmaceutical agents are being developed: Radiolabeled ICI like 89Zr-nivolumab or 89Zr-pembrolizumab as well as agents targeting interleukin-2 could aid in the selection of patients who will benefit from ICI-therapy [18]. Data on these substances are still limited to animal models and early-phase human studies; however, an initial trial using ^89^Zr-atezolizumab for PET imaging showed very promising results assessing clinical response to PD-L1 blockade in 25 patients with locally advanced or metastatic bladder cancer, non-small-cell lung cancer (NSCLC) or triple-negative breast cancer. Interestingly, responses to atezolizumab therapy in these patients had a better correlation with the pretreatment ^89^Zr-atezolizumab PET signal than with immunohistochemistry- or RNA-sequencing-based predictive markers [44]. Further clinical studies are ongoing and can hopefully be expected in the future.

A very interesting novel approach is based on the use of Zr-89-labeled minibodies that target CD8-positive T cells [45]. Since tumor-infiltrating T cells, in particular CD8-positive T cells, play an important role for initiating and mediating a response to ICI, the in vivo visualization of CD8-positive T-cell-rich tissue might be crucial for development of more effective ICI therapies. A subsequent clinical phase 2 study using ^89^Zr-IAB22M2C for PET-CT imaging is ongoing among patients with Melanoma, Non-Small-Cell Lung Cancer, Renal Cell Carcinoma, and Squamous Cell Carcinoma of the Head and Neck, aiming to predict response to ICI (NCT03802123).

Another novel promising imaging approach was reported by Chatterjee et al., who developed highly specific radiolabeled peptides for PET imaging of PD-L1 tumor expression [46,47,48]. Preclinical studies suggest excellent imaging properties and the potential to significantly influence the standard clinical workflow in ICI-treated patients, as these radiopharmaceuticals can be used not only for the therapy guidance and monitoring but also for optimizing dose and therapeutic regimes in a very individual way [48]. Initial clinical trials are ongoing and results may be expected in the near future. 

## 4. Current Clinical Applications and Research Advances

### 4.1. Thoracic Tumors

According to current clinical practice guidelines, PET/CT is recommended in the staging process of both small-cell lung cancer (SCLC) and non-small-cell lung cancer (NSCLC), as it can improve diagnostic accuracy as compared to conventional staging using CT [49,50,51,52,53]. This is especially relevant in the early and locally advanced setting for the determination of operability, planning of radiotherapy and in case of suspected recurrence after therapy with curative intent [49,51]. In stage IV disease, PET/CT is recommended for staging, especially for the detection of bone metastases and for assessing biopsy sites [51]. Its application in pleural mesothelioma or thymic tumors is not routinely recommended; however, PET/CT may be of use when routine staging is unclear, for assessing biopsy sites or to characterize lesion activity in suspected recurrent tumors [54,55].

General limitations of PET/CT in thoracic tumors are its low sensitivity in small lesions, movement artifacts due to breathing and the possibility of false-positive results due to inflammatory processes like in infectious or inflammatory diseases, after surgery, radiotherapy or pleurodesis. False negative findings may also occur in tumors with low cell density and low PET avidity, e.g., in bronchoalveolar carcinoma [51,52,54].

Despite the fact that current guidelines do not routinely recommend PET/CT for follow-up of any of the mentioned thoracic tumor entities [49,50,51], evidence on the utility of PET/CT in assessing response to ICI therapies, especially for NSCLC, is evolving: Tumor SUVmax has been excluded as a significant predictor of survival in different analyses [10,56,57], but it correlates with PD-L1 status [10,58,59], and thus also with response to ICI therapies [60]. As SUVmax has the major limitation of its dependence on tumor size, volume-based parameters should be primarily applied [61]. For NSCLC patients undergoing chemotherapy, tumor as well as whole-body (wb) tumor lesion metabolic tumor volume (MTV) and total lesion glycolysis (TLG) were significantly predictive of overall survival (OS) [56]. Similar data could also be shown for patients undergoing PD-1/PD-L1-directed ICI therapy for previously treated NSCLC [57].

This concept was recently refined by Jreige et al., who established the ratio of metabolic-to-morphological lesion volumes (MMVR), which delineates the fraction of necrosis within a tumor lesion. MMVR was found to be negatively correlated to PD-L1 status, and low MMVR was associated with better survival rate. In line with the imaging aspect derived from PET/CT, a retrospective evaluation of histological samples revealed that tumors with necrosis showed a greater frequency of CD8+ lymphocytes [42]. This corresponds to the hypothesis that rather “inflammatory” or immunologically “hot” tumors, presenting with an extensive (and PET-active) immune cell infiltrate, may show better response to ICI therapy [1,62]. Such immune responses may go along with the formation of necrosis within the tumor, whereas PET/CT may be of special value helping to distinguish necrosis from vital tumor tissue in clinical practice.

A major field of scientific advance is the combination of PET/CT-derived biomarkers with other patient- or tumor-related biomarkers: Recently published data demonstrated that a composite biomarker combining the PET/CT-derived variable total (T) MTV together with the derived neutrophil-to-lymphocyte ratio (DNLR) was highly predictive of OS in NSCLC patients treated with ICI [10]. On the other hand, data suggest that patients with high MTV and increased DNLR are at higher risk for hyperprogressive disease upon ICI therapy [63].

Another recently reported approach is the assessment of PET/CT early after ICI initiation: Kaira et al. showed that uptake dynamics after 1 month of nivolumab therapy in NSCLC patients were a better predictor of the further course of disease than corresponding CT measures [12]. Similar approaches also have been reported for composite biomarkers including PET/CT parameters: Castello et al. reported on a “immune-metabolic-prognostic index (IMPI)” of TLG < 541.5 mL and NLR < 4.9 measured at 8 weeks after ICI initiation that could stratify the risk of further progression or survival [64].

In another noteworthy study, Humbert et al. investigated NSCLC patients under therapy with pembrolizumab or nivolumab, who had PD according to PERCIST at an interim PET/CT at week 7 after treatment initiation, but without clinical worsening. In those patients, therapy was continued until another PET/CT evaluation after 3 months. Of 19 patients with initial PD but ICI treatment being continued, 42% had a homogeneous disease progression and did not reach a durable clinical benefit (DCB), defined as ICI continuation of six months. In addition, 26% had immune-dissociated response and 32% had pseudoprogression, both categories leading to a DCB [27]. This suggests that longitudinal follow-up with PET/CT may identify atypical response patterns more frequently than expected, which could allow for better stratification and prevention of premature ICI therapy withdrawal. 

Of interest, PET/CT may also have implications for predicting the development of IRAE: Hashimoto et al. showed that the frequency of IRAE was higher in patients with low tumoral FDG uptake [57]. Data from a small patient cohort reported by Eshghi et al. suggested that ^18^F-FDG uptake in the thyroid gland under therapy with ICI in lung cancer patients can predict thyroiditis even before laboratory testing [65]. Our own clinical practice suggests that IRAE are quite frequently detectable in PET/CT, as shown in Figure 2 and Figure 3. At the moment however, such findings are usually discovered by chance rather than PET/CT would be applied to screen for them. 

### 4.2. Dermatooncology

For staging and therapy monitoring of patients with advanced melanoma undergoing immunotherapy, PET/CT is increasingly used in clinical practice. Due to the possibility of atypical response patterns previously mentioned, classical imaging response criteria such as RECIST have been shown to frequently over- or underestimate the therapeutic effect of ICI [66]. Hodi et al. showed, e.g., that evaluating the success of therapy with Pembrolizumab in metastatic melanoma patients with RECIST underestimates the benefit in approximately 15% of individuals [67]. As a result, criteria such as *immune-related response criteria* (irRC), PERCIMT or iRECIST have been established [66].

Despite the challenges that come with evaluating ICI therapy response with ^18^F-FDG-PET/CT in melanoma patients, potential new benefits of this imaging technique arose recently: multiple authors suggest PET/CT as a prognostic tool, as pre-treatment- or therapy-monitoring-examinations may provide information regarding future therapy response [68]. Seban et al., for example, proclaimed the value of quantitative baseline ^18^F-FDG PET/CT biomarkers as a prognostic tool in patients with cutaneous and mucosal melanoma. Their data suggested a negative correlation between high baseline metabolic tumor burden or bone marrow metabolism and ICI response for cutaneous melanoma. On the other hand, for mucosal melanoma, high baseline SUVmax was found to be the only prognostic baseline imaging biomarker [69]. 

Partly based on these data, Hindié et al. recently suggested that patients with high tumor burden as defined by total metabolic tumor volume (TMTV) in baseline PET/CT were unlikely to respond to ICI monotherapy. For these patients, alternative therapies such as BRAF/MEK inhibitor therapy or combined ICI therapy might be a better primary therapy option. Furthermore, CR in 18F-FDG PET was discussed as a strong indicator for safe discontinuation of ICI therapy [68].

Nakamoto et al. investigated the prognostic value of wbMTV three months following initiation of ICI and concluded that it was a strong indicator of OS in advanced melanoma patients. They further stated that the possibility of pseudoprogression must be considered during early restaging images but noted that it had only occurred in one of their eighty-five patients evaluated [6].

Another promising feature of ^18^F-FDG PET/CT scans was suggested by Wong et al., who discovered that a pre-treatment spleen-to-liver ratio of >1.1 was associated with poor outcome after ipilimumab treatment in advanced melanoma. This could not be shown for patients treated with PD-1 antibodies, but the authors criticized the small sample size of the anti-PD-1 cohort. Further investigation of this ^18^F-FDG-PET/CT signature with larger cohorts of patients treated with PD-1 antibodies or combined ICI was recommended [70].

Very recently, a first systematic review and metanalysis concerning prediction and monitoring of immunotherapy response in patients with metastatic melanoma was performed by Ayati et al. The authors concluded that baseline metabolic tumor burden parameters such as SUVpeak, MTV and TLG were promising biomarkers for response prediction and that image evaluation using modified response assessment criteria such as PERCIMT was generally regarded as superior compared to conventional criteria in most applications. Finally the authors stated that the occurrence of particularly severe IRAE was correlated with response, but was not essential for clinical benefit [71].

Besides its role as a follow-up—and potentially prognostic—tool, recent data suggest that relevant IRAE can be detected by ^18^F-FDG-PET/CT, possibly even ahead of clinical diagnosis [72], whereas Mekki et al. reported an IRAE detection rate of 83% with ^18^F-FDG-PET/CT. Interestingly, detection rates varied distinctly for various organs. Further investigation regarding this topic is advisable because of low patient numbers in this study [73]. In our clinical practice, as already reported for thoracic tumors, visualization of IRAE rather happens by chance, but such findings can be clinically relevant, as described in Figure 4.

### 4.3. Urological Malignancies

There are limited data regarding the significance of PET/CT in ICI therapy of urothelial cancer. A case report of a patient with lynch syndrome who underwent sequential immunotherapy with atezolizumab, pembrolizumab and ipilimumab/nivolumab for urothelial cancer of the bladder and ureters, which occurred alongside a recurrence of colorectal carcinoma, showed a metabolic response on 18F-FDG PET/CT [74]. In a murine bladder cancer model, tumor-infiltrating T cells could be visually quantified in PET by way of an anti-CD3 monoclonal antibody, which was modified with desferrioxamine and radiolabeled with zirconium-89 [75].

Due to the high activity of FDG in 18F-FDG, PET/CT has shown disappointing results in the differentiation of renal masses, with a pooled sensitivity of 50–60% [76]. Its poor diagnostic performance stems from a high rate of false negatives when compared to conventional CT imaging, which has been shown in several prospective and retrospective trials [77,78]. This prompted guidelines to recommend against using 18F-FDG PET/CT in staging and surveillance of renal cancer [79]. However, it shows excellent specificity and sensitivity in the diagnosis of metastatic lesions of renal cancer [80].

While there seems to be no added benefit of 18F-FDG PET/CT in comparison with conventional CT in primary diagnosis and initial staging of renal malignancies, beneficial results have been shown in the evaluation of treatment response. High SUVmax correlated with high-grade, poor prognosis and prediction of metastatic disease [81,82,83]. Lately, tracers that are more specific to renal cell carcinoma than FDG have been developed, such as 89Zr-girentuximab [84].

In prostate cancer, no ICI substance has yet shown efficacy sufficient for approval, but trials are ongoing, [85] and preliminary data suggest durable response to ICI combination therapies in a highly selected subset of pretreated patients [86,87,88,89]. In this light, prostate-specific membrane antigen (PSMA) PET/CT warrants mention. This highly specific ^68^Ga- and ^18^F-bound tracer substance provides excellent contrast-to-noise ratio and only suffers from few false positives like other non-prostatic malignancies, sarcoidosis or benign bone diseases. In addition to computed tomography, a recent systematic review has shown superior diagnostic accuracy for PSMA-PET/MRI when compared to the current gold standard, multiparametric magnetic resonance imaging (mpMRI) [90]. There is increasing evidence that PSMA imaging influences clinical decision-making [91]. Recently, advances have been made to establish it in primary staging of high-risk disease, where a systematic review and meta-analysis attested ^68^Ga-PSMA PET/CT a pooled sensitivity and specificity for lymph node involvement before treatment of 75% and 99%, respectively [92]. In the wake of possible future ICI therapies for prostate cancer, however, the significance of PSMA-PET/CT remains to be investigated.

### 4.4. Hemato-Oncology

Staging with PET/CT is critically important for patients with aggressive hemato-oncological diseases such as lymphomas. Precise staging before therapy initiation reduces the risk of overtreatment and consecutive toxicity due to extended-field radiation or prolonged chemotherapy [93].

Despite the well-established role of PET-imaging at baseline and under therapy, surveillance imaging is discussed controversially in literature, as relapse is mostly detected by clinical examination or the patients’ awareness.

With novel immunotherapy options like ICI, Bi-specific T-Cell Engagers (BiTEs) and Chimeric Antigen Receptor (CAR-) T cells having emerged especially in the therapy of aggressive lymphomas, PET/CT for staging, and response evaluation is gaining increasing impact based on a growing body of evidence.

Concerning ICI, the value of early PET/CT two months after nivolumab initiation as compared to regular CT imaging in relapsed or refractory Hodgkin Lymphoma has been recently analyzed retrospectively by Mokrane et al. Both PET/CT and CT response criteria were correlated with OS, but PET/CT could identify more patients with complete responses [94].

BiTEs are novel bispecific antibodies or antibody fragments connecting T cells and tumor cells directly, binding CD3 on the T cell and CD19 on the lymphoma cell, enabling the malignant cell to be attacked. In a 2016 phase 2 study on blinatumomab by Viardot et al., PET/CT and the “2007 revised response criteria” were used for the first time in such a therapeutic setting. Three patients who had a CT-defined PR that was confirmed by PET at week 10 later achieved CR in the follow-up period (3–12 months), indicating a prolonged immune response or a lag time in the demonstration of CR [95]. These observations suggest that timing and value of PET/CT scans after immunotherapy still need to be defined.

CAR-T cell therapy is based on patient T cells being genetically reprogrammed. These cells synthesize a protein on the T-cell surface binding to the surface proteins of the tumor cell (e.g., CD19 or CD28), which then acts as a reprogrammed killer T cell after binding to the tumor cell. 

Usage of PET/CT in patients treated with CAR-T cells has recently been examined by Shah et al. in seven patients with diffuse large B-cell lymphoma (DLBCL). In PET/CT one-month post-treatment, three patients concurrently showed no sign of MTV activity, two had an increase in MTV and two had residual MTV. Two years post-treatment, only those three patients without residual MTV remained in CR. Development of cytokine release syndrome (CRS) did not confound PET/CT findings [96].

### 4.5. Head and Neck Cancer

PET/CT has become an important tool in the evaluation of head and neck cancer (HNC) in the last years. Especially in advanced stages, clinicians benefit from higher sensitivity regarding stage migration [97]. Relapses in the first three years are frequently encountered and especially recurrent HNC is considered to be of poor prognosis [98,99,100]. Since in these cases curatively intended surgery is extensive and significantly reduces the quality of life, a careful evaluation of the therapeutic intent is imperative [101,102]. A prospective, randomized, controlled trial could show that PET surveillance could reduce neck dissection after chemoradiotherapy by 80% without impact on OS but distinct cost reduction [103]. Due to the detection of distant metastases, Rohde et al. found a significant shift towards palliative treatment decisions of multidisciplinary tumor conferences in recurrent head and neck squamous cell carcinomas (HNSCC) using PET/CT compared to MRI and chest X-ray [102].

Summarized data analyzing PET/CT in the evaluation of ICI therapy in HNC is not yet available. However, one promising clinical PD1/-L1 imaging study is upcoming with the PINCH trial (NCT03829007; PINCH – PD-L1 ImagiNg to predict Durvalumab Treatment Response in HNSCC), measuring the uptake of ^89^Zr-durvalumab in 58 participants with relapsed or metastatic HNSCC [104]. Data on the potential to predict disease control rate and on the correlation between tracer uptake and PD-L1 expression determined that immunohistochemically is awaited to be published in 2021.

## 5. Current Challenges, Future Prospects and Closing Remarks

In synopsis, ^18^F-FDG PET/CT currently plays an important role in staging and follow-up of FDG-avid tumor entities, like NSCLC, melanoma or aggressive lymphomas. However, PET/CT follow-up of patients under antineoplastic therapy is not routinely recommended in most situations. With ICI therapy rapidly evolving in numerous tumor entities, and due to the challenges associated with assessing radiological response to such therapies, there is an urgent clinical need for additional imaging-derived biomarkers. The large number of studies on PET/CT biomarkers in ICI therapy response evaluation featured in our review reflects this quest and testifies that there already is a growing body of evidence for such metabolic imaging strategies. Biomarkers derived from PET/CT are promising for the development of personalized treatment strategies, especially concerning the choice of therapy and for its longitudinal management. Their potential prognostic appliance is especially appealing considering the current lack of established highly predictive blood- or tumor-related biomarkers.

Inherent technical limitations of ^18^F-FDG-PET/CT have been previously discussed and include false-positive findings in inflammatory lesions, tumor entities showing no or little FDG uptake or physiological uptake like in the urinary tract. It is likely that these challenges associated with the low specificity of FDG uptake can be overcome with novel, more specific, tracer substances that are currently under development. Using tracers linked to the respective ICI substance, actual “theranostic” approaches may become possible, including more specific pre-therapy response prognosis and follow-up assessment. Also, the idea of non-invasive estimation of immunohistochemical features like PD-L1 expression using special tracers appears highly attractive.

Another limiting factor to be considered in the application of PET/CT in ICI-treated patients is the obvious shortage of PET/CT resources in some regions [105]. Also, in several malignancies, studies on the use of PET/CT clearly have not been able to keep up with the pace of the rapidly evolving immunotherapy landscape: while, for example, most data on PET/CT in ICI therapy of NSCLC are based on retrospective mono-immunotherapy cohorts, ICI therapy has broadly entered first-line therapy, usually in combination with chemotherapy [106,107,108]. Also, in NSCLC, ICI substances are increasingly used in locally advanced disease stages [109]. For both settings, data on PET/CT and response assessment or even response prediction are not available yet.

Reviewing the large number of studies published concerning PET/CT for response assessment to ICI therapy, we note that, to date, there still is a striking discrepancy with the relevant clinical practice guidelines which, across the board, do not suggest using PET/CT in this indication. On the one hand, this is due to the extensive lack of prospective data allowing to draw conclusions as to whether there is a clinically significant benefit of using PET/CT versus conventional imaging in such therapy settings. On the other hand, PET/CT imaging and various derived biomarkers frequently lack standardization, which precludes their broader implementation in clinical practice. Interim PET/CT examinations could aid an early response assessment and help to identify atypical response patterns, especially hyperprogression or pseudoprogression. Standardized response criteria like PERCIST may be helpful in that field; however, we assume that also in future they will rather be applied in clinical trials than in daily clinical practice as they do not have immediate implications on the individual patient’s level. Also, we believe it is likely that, rather than singular PET-derived biomarkers, the combination of a broader panel also including blood and histological parameters will be necessary for a significant clinical impact.

Due to these challenges, it is still unclear how PET/CT can be embedded in clinical decision-making apart from staging and re-staging. We believe that there is large future potential, especially for pre-ICI therapy PET/CT biomarkers to stratify patients for maximizing the therapeutic effect of ICI therapy. However, unless high-quality clinical trials demonstrate a significant survival or response benefit and thereby also define the methods to be used in future, PET/CT for ICI therapy response assessment is not yet ready for broader application. Right now, it is time to focus the numerous existing pieces of evidence, to standardize technique, sequencing and biomarkers and to translate that knowledge into high-quality clinical trials. Only by doing so, PET/CT assessment in ICI therapy can finally be implemented in daily clinical practice based on a solid body of evidence.

## Figures and Tables

**Figure 1 jcm-09-03483-f001:**
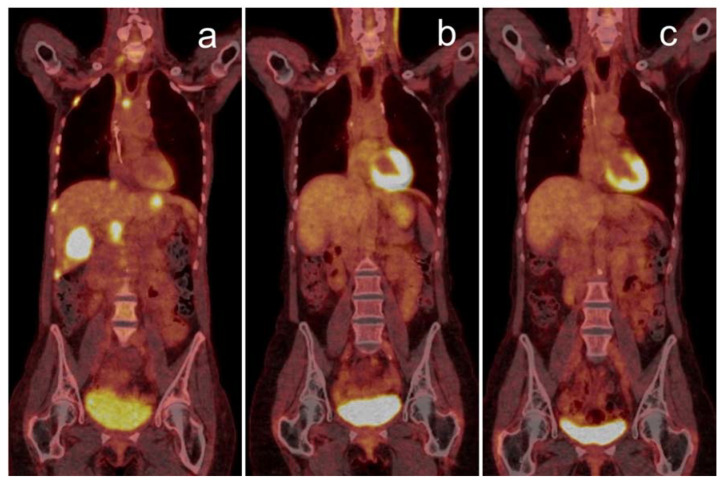
Complete metabolic response over two years visualized by PET/CT: 68-year-old woman with stage IV lung squamous-cell carcinoma (PD-L1 1%) having progressed after first-line chemotherapy, showing multiple liver, bone and lymph node metastases (**a**). The patient responded well to pembrolizumab monotherapy, with a PET/CT follow-up scan after one year (**b**) showing complete metabolic remission. Therapy was completed after two years, still in complete metabolic remission (**c**). PET/CT: positron emission tomography/computed tomography; PD-L1: programmed death-ligand.

**Figure 2 jcm-09-03483-f002:**
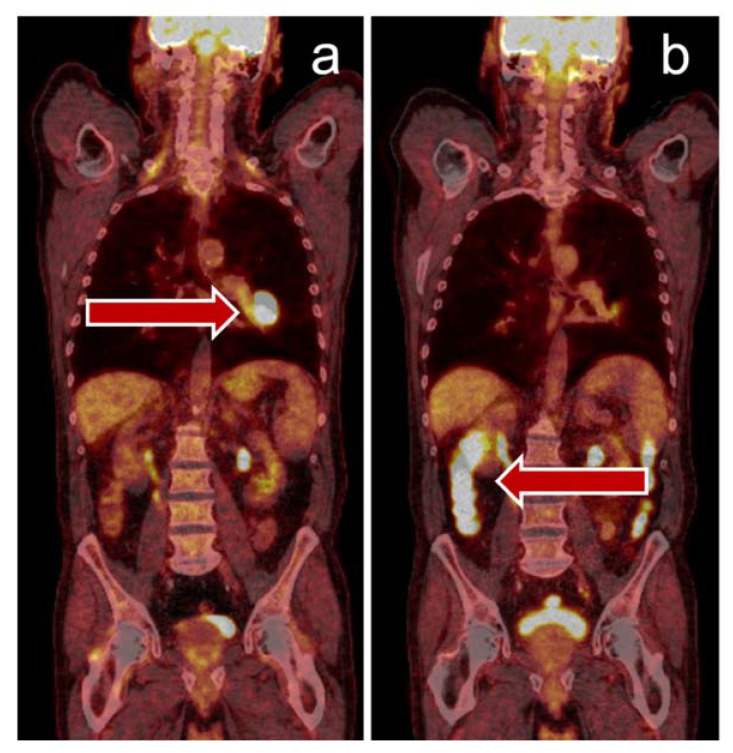
Immune-related colitis visualized by PET/CT: 71-year-old man with stage IV NSCLC, left hilum (PD-L1 90%) with cervical lymph node metastasis that was surgically removed for histological analysis (**a**). After three cycles of pembrolizumab, the patient underwent a follow-up PET/CT scan for evaluation of possible radiotherapy and reported persistent diarrhea. PET/CT scan showed a major tumor response and increased uptake in the colon region consistent with immune-mediated colitis (**b**). Colonoscopy and histological specimens supported this diagnosis, the patient responded to corticosteroid treatment, immunotherapy was paused. NSCLCL non-small-cell lung cancer.

**Figure 3 jcm-09-03483-f003:**
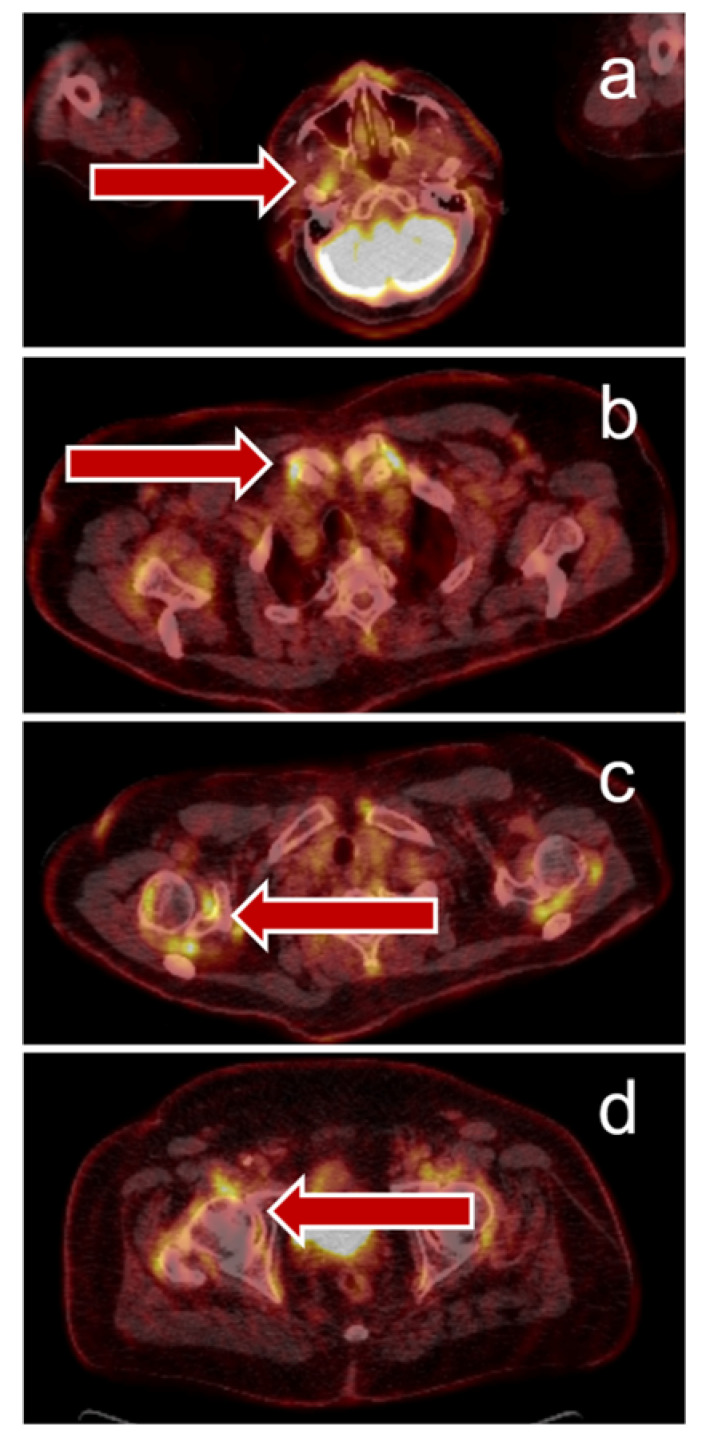
Immune-related arthritis visualized by PET/CT: 69-year-old man with stage IV NSCLC (PD-L1 100%), small left hilar tumor, adrenal metastasis, and symptomatic brain metastasis – resected). The patient reported a history of chronic polyarthritis but did not require therapy at time of cancer diagnosis. The patient received six cycles of pembrolizumab monotherapy and reached partial remission. Due to increasing joint pain and suspected bone metastases, PET/CT was performed and showed a complete metabolic tumor remission, but hypermetabolic joint lesions ((**a**)–temporomandibular joints, (**b**)—sternoclavicular joints, (**c**)—shoulders, (**d**)—hips) compatible with immune-mediated arthritis. The patient received corticosteroid therapy and immunotherapy was paused, the joint pain improved.

**Figure 4 jcm-09-03483-f004:**
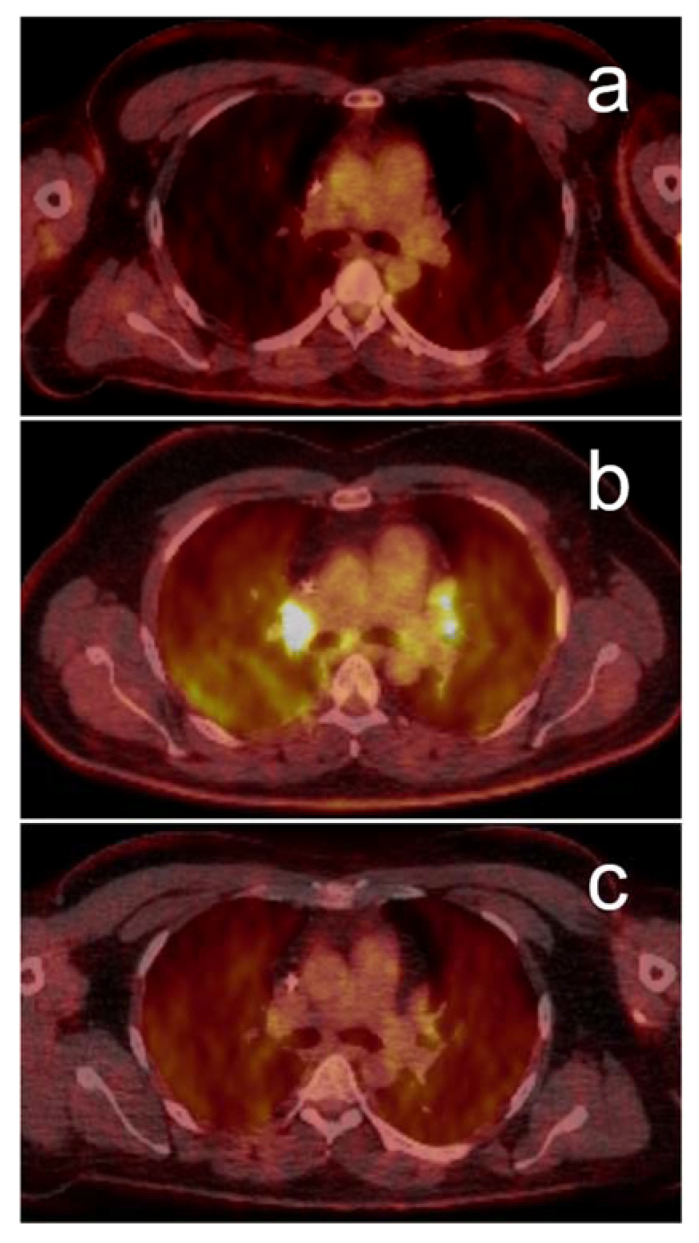
Suspected immune-related sarcoid-like reaction visualized by PET/CT: 37-year-old man with recurrent metastatic melanoma, a single lung metastasis was surgically removed for diagnostic reasons before treatment with pembrolizumab was initiated. Pre-therapy PET/CT (**a**) shows no evidence of intrathoracic activity. Clinically and in conventional imaging, a complete remission was reached within few months, but a subsequent follow-up PET/CT six months after therapy initiation (**b**) showed bilateral mediastinal and hilar lymphadenopathy and diffuse lung parenchymal uptake. As the patient did not report any respiratory or tumor-specific symptoms, a sarcoid-like reaction was suspected based on PET/CT morphology. Pembrolizumab was resumed, but a timely PET/CT surveillance scan after three months was scheduled, which showed spontaneous resolution of the hypermetabolic lesions (**c**).

**Table 1 jcm-09-03483-t001:** Summary of quantitative biomarkers that can be derived from PET/CT imaging.

SUV (standardized uptake value)	Quantitatively describes the glucose metabolism of a lesion. Regional radioactivity concentrations, determined by the dose administered, the decay of the nuclide and patient’s weight.
SUVmax (maximum standardized uptake value)	Represents the most intensive ^18^F-FDG uptake in the tumor, maximum SUV value of a region based on a single voxel value only. Often used as a parameter for nuclide uptake, but may be misleading, as it represents only a single voxel value. Thus, it is susceptible to noise, dependent on image resolution, and on the voxel of interest (VOI) definition [4]. An advantage of SUVmax is that placement of the VOI is not critical.
SULpeak (standardized uptake value corrected for lean body mass)	Measured in a 1 cm^3^ volume around the hottest voxel in the tumor. Is considered a more stable alternative to the noise-susceptible measurement of the SUVmax [5].
MTV (metabolic tumor volume) *	Represents the volume of a tumor lesion with increased ^18^F-FDG uptake. Whole-body (wb) or total (T) MTV has been defined as the sum of the individual MTVs of all lesions with SUV ≥ 2.5 [6] and has been shown to be a particularly strong prognostic factor in pre-ICI treatment melanoma and NSCLC patients [6,7,8,9,10]. Concerning early response assessment in NSCLC, the increase in wbMTV six weeks after ICI initiation indicated poorer outcomes even in the case of stable disease by CT assessment [11].
TLG (total lesion glycolysis) *	Defined as the product of the MTV and the mean SUV, integrating the tumor-related metabolic activity and tumor volume. In contrast to the SUV, it does not describe the maximum or average glucose turnover at a specific point, but rather the glucose turnover of all lesions. Metabolic tumor response as assessed by TLG may be a more precise predictor of prognosis than MTV or SUVmax [12].

* In contrast to the estimation of SUVmax, determination of MTV and TLG depends on the placement of the VOI. Several approaches can be applied and need to be specified in the methods of the respective publication. Another practical problem with MTV and TLG is that it can be difficult or even impossible to apply in the case of a large number of metastatic lesions, since a VOI has to be created for each individual lesion. ICI: immune-checkpoint inhibitors; NSCLC: non-small-cell lung cancer.

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
