# Peer review of "Impact of PET/CT for Assessing Response to Immunotherapy—A Clinical Perspective"

_jcm, 2020, doi:10.3390/jcm9113483_

Round 1

Reviewer 1 Report

Overall, this paper is an excellent review of the current status of PET/CT imaging in the assessment of immune checkpoint inhibitor therapy.  There are a few gaps that should be considered; see below.

An important general point that should be made in the introduction is that throughout the paper PET/CT refers to FDG PET/CT unless stated other wise.  It could be done in a short paragraph analogous to the last paragraph in the introduction.

Table 1:

An advantage of SUVmax is that placement of the VOI (not ROI, which is for planar images) is not critical. All the operators will get the same value as long as they are looking at the same lesion. In contrast, determination of MTV and TLG are very dependent on VOI placement. There are several approaches that can be used and must be specified in the methods section of a paper. Another practical problem with MTV and TLG is that it is very difficult or impossible to do if there are a large number of metastatic lesions, since a VOI has to be created for each one.

Another general point about quantitative indices (including textural analysis) is that they are all dependent on the quality of the PET images.  Very different results are obtained on a 20 year-old machine vs. a new one.  Not only that, but almost everyone uses a different reconstruction methodology. This makes it quite difficult to apply any quantitative measures done at one institution to another. Careful standardization is essential to use these methods.

See Sunderland JJ, Christian PE. Quantitative PET/CT scanner performance characterization based upon the society of nuclear medicine and molecular imaging clinical trials network oncology clinical simulator phantom. J Nucl Med. 2015 Jan;56(1):145-52. and Graham MM, Badawi RD, Wahl RL. Variations in PET/CT methodology for oncologic imaging at U.S. academic medical centers: an imaging response assessment team survey. J Nucl Med. 2011 Feb;52(2):311-7.

In the section “Novel approaches…”

One approach that should probably be mentioned is the use of Zr-89 labeled minibodies that target CD8. See Pandit-Taskar N, et al. First-in-Humans Imaging with (89)Zr-Df-IAB22M2C Anti-CD8 Minibody in Patients with Solid Malignancies: Preliminary Pharmacokinetics, Biodistribution, and Lesion Targeting. J Nucl Med. 2020 Apr;61(4):512-519.

 A phase 2 study with this agent is underway looking at T-cell recruitment into tumors after administration of an ICI in patients with Melanoma, Non-Small Cell Lung Cancer, Renal Cell Carcinoma, and Squamous Cell Carcinoma of the Head and Neck. The goal is to predict responders vs. non-responders. See trial NCT03802123 at clinicaltrials.gov.

Line 182-183: “Radiolabeled agents containing ICI substances like nivolumab” would be clearer as “Radiolabeled ICIs like 89Zr-nivolumab or 89Zr-pembrolizumab….”

Line 220 should probably be “standard of care”.

Line 469 Rather than “Due to the physiological functional overlap in glucose metabolism …”, the problem should be stated as “Due to the high activity of FDG in urine …”

In the section on urothelial cancers you should consider at least brief mention of the PSMA radiotracers. See: Evangelista L, et al.  PET/MRI in prostate cancer: a systematic review and meta-analysis.  Eur J Nucl Med Mol Imaging. 2020 Sep 8. Online ahead of print.

These are likely to result in significant changes in the management of prostate cancer patients and are likely to be useful if ICI therapy is used.  See: Patel D, McKay R, Parsons JK. Immunotherapy for Localized Prostate Cancer: The Next Frontier? Urol Clin North Am. 2020 Nov;47(4):443-456.

Reviewer 2 Report

The manuscript is a narrative review that discusses response assessment and pretherapy response prediction in ICI therapy in various cancers. The aims are to give a clinical overview on present standards and evidence and to identify current challenges and future perspectives.

The topics are addressed within different fields: Nuclear medicine, thoracic tumors, dermatology, hemato-oncology, urological malignancies, and head and neck cancer. Wonder why gastrointestinal, breast, sarcoma, and gynaecological cancers are not included in the review.

The title of the manuscript is interesting and the aims are good. I do not have expertise within all these fields and was therefore excited to start reading. This turned however out to be challenging as there is too little real information to truly understand on one hand, and too complicated and very condensed information on the other hand. This resulted in a disappointing experience. 

Basically, the manuscript is six small manuscripts put together into one. Each little manuscript addresses the current application of PET/CT for a given tumor. It is very condensed and, does not add anything to the well informed reader specialised in one or more of the cancers and to not explain enough for those who are not sepcialists. Then follows the interesting part – “Resent advances in research on PET/CT and ICI..” This part should be occupying the majority of the paper and be explained much more extensively to be informative and understandable. To bring something new/interesting the results should be put together across tumor types and discussed in an "immunologic perspective".  Finally I would like the authors to take position: There is not and never will be enough evidence for any diagnostic imaging procedure but do the authors think that is PET a valuable tool?  What should the modified criteria capture? Is it worth developing/refining these or more new criteria/predictors or dose it only add to confusion and reduce comparability between studies? Do the authors have recommendation regarding response criteria/predictors that should always be included/reported in research studies?

Each little manuscript has an introduction and some final remarks regarding future perspectives. I tried to read all the introductions in continuity and all the final remarks in continuity and the similarity was striking. This explains why the manuscript is kind of boring. 

The nuclear medicine perspective

The introduction (lines 99-141) is very general and could in my opinion be reduced to few lines or omitted. Otherwise it should be more extensive and explain the problems with SUV measurements and take the discussion from the technical level (SUVmax vs SUVpeak, TLG, MTV etc) to a cellular/tissue level describing the changes in tumor microenviroment during ICI treatment that causes false positive findings.  SULpeak should be included in table 1 as it is part of the modified PECRIT criteria mentioned later in the manuscript.

The description of the typical response patterns is OK. However the durable responses (even after termination of therapy, PR, SD and CR) should also be addressed. Think it would be more interesting to replace figure 1 with a case illustrating an atypical response - preferable pseudoprogression.

The modified immune response criteria iPERCIST, PERCIMT, PERCIST and imPERCIST are mentioned. A more extensive description would add further to the understanding – a Table 2, maybe? This information is however also published elsewhere. The only systematic review and meta-analysis of the value of FDG PET/CT for predicting or monitoring immunotherapy response in patients with metastatic melanoma should be mentioned (The value of (18)F-FDG PET/CT for predicting or monitoring immunotherapy response in patients with metastatic melanoma: a systematic review and meta-analysis. Ayati N, Sadeghi R, Kiamanesh Z, Lee ST, Zakavi SR, Scott AM.Eur J Nucl Med Mol Imaging. 2020 Jul 29. doi: 10.1007/s00259-020-04967-9. Online ahead of print).

Regarding novel approaches textural features should be shortly explained – especially as the reference is seems to be in German language (unfortunately I do not have access to the journal). Line 208-212 can be omitted.

Thoracic tumors, dermatology, hemato-oncology, urological malignancies, and head and neck cancer. Regarding urothelial cancer and renal cancer data in the literature are very limited and these sections of the paper are therefore not informative other than PET/CT is not used for these cancers. Regarding tracers for renal cell cancer PSMA which is widely available should be mentioned.
